# Bacterial communities of *Aedes aegypti* mosquitoes differ between crop and midgut tissues

**Luis E. Martinez Villegas**[1☯], **James Radl**[1☯], **George Dimopoulos**[2], **Sarah M. Short**[1,2]*

**1** Department of Entomology, The Ohio State University, Columbus, Ohio, United States of America,
**2** Department of Molecular Microbiology and Immunology, Johns Hopkins Bloomberg School of Public Health, Baltimore, Maryland, United States of America

☯ These authors contributed equally to this work.
* short.343@osu.edu

**Data Availability Statement:** Raw data is contained within Supplementary File 1, with the exception of raw sequence reads from high throughput amplicon sequencing. These

## Abstract

Microbiota studies of *Aedes aegypti* and other mosquitoes generally focus on the bacterial communities found in adult female midguts. However, other compartments of the digestive tract maintain communities of bacteria which remain almost entirely unstudied. For example, the Dipteran crop is a food storage organ, but few studies have looked at the microbiome of crops in mosquitoes, and only a single previous study has investigated the crop in *Ae. aegypti*. In this study, we used both culture-dependent and culture-independent methods to compare the bacterial communities in midguts and crops of laboratory reared *Ae. aegypti*. Both methods revealed a trend towards higher abundance, but also higher variability, of bacteria in the midgut than the crop. When present, bacteria from the genus *Elizabethkingia* (family Weeksellaceae) dominated midgut bacterial communities. In crops, we found a higher diversity of bacteria, and these communities were generally dominated by acetic acid bacteria (family Acetobacteraceae) from the genera *Tanticharoenia* and *Asaia*. These three taxa drove significant community structure differences between the tissues. We used FAPROTAX to predict the metabolic functions of these communities and found that crop bacterial communities were significantly more likely to contain bacteria capable of methanol oxidation and methylotrophy. Both the presence of acetic acid bacteria (which commonly catabolize sugar to produce acetic acid) and the functional profile that includes methanol oxidation (which is correlated with bacteria found with natural sources like nectar) may relate to the presence of sugar, which is stored in the mosquito crop. A better understanding of what bacteria are present in the digestive tract of mosquitoes and how these communities assemble will inform how the microbiota impacts mosquito physiology and the full spectrum of functions provided by the microbiota. It may also facilitate better methods of engineering the mosquito microbiome for vector control or prevention of disease transmission.

sequences can be found in the Sequence Read Archive under BioProject PRJNA915094.

**Funding:** This work was funded by the National Institutes of Health, National Institute for Allergy and Infectious Disease (https://www.niaid.nih.gov/), Grant R21AI136456 and the Bloomberg Philanthropies (https://www.bloomberg.org/) to GD, as well as the Ohio State University Infectious Diseases Institute (https://idi.osu.edu/) and the Ohio State University College of Food, Agricultural, and Environmental Sciences (https://cfaes.osu.edu/) to SMS. SMS was also supported by a Ruth L. Kirschstein National Research Service Award F32AI112208-01A1. The funders had no role in study design, data collection and analysis, decision to publish, or preparation of the manuscript.

**Competing interests:** The authors have declared that no competing interests exist.

## Author summary

Bacteria inside mosquitoes' guts have been found to have an impact on mosquito life history traits (such as longevity and fecundity) as well as their susceptibility to infection by human pathogens. Engineering these communities may provide an effective and safe way to control mosquitoes and reduce the impact of the pathogens they spread. In this work, we assayed the bacteria found in midgut and crop tissues of a medically important mosquito, *Aedes aegypti*. Our results show that these tissues harbor communities of bacteria that differ in composition and function and vary in abundance. Experiments like ours are important to better understand where bacteria are found in an insect's body and how these communities assemble. This knowledge may help future researchers more successfully engineer bacterial communities in mosquitoes.

## Introduction

The yellow fever mosquito, *Aedes aegypti* (L.), transmits multiple human arboviral pathogens (including dengue virus, Zika virus, chikungunya virus, and yellow fever virus) that collectively cause more than one hundred million human infections each year and tens of thousands of deaths [1]. Vaccines exist for some of these arboviruses with varying efficacy [2,3]. However, many lack vaccines, and treatment is limited in all cases to supportive care. Because arboviral threats are often emerging and challenging to predict [4], control methods targeting mosquito vectors are often the primary mode to prevent transmission and reduce disease cases.

*Ae. aegypti* mosquitoes are found in close association with microbial communities throughout their lives. The most well-studied of these associations are the bacterial communities that inhabit the larval habitat and the larval and adult female digestive tract. As larvae, *Ae. aegypti* develop in stagnant water found primarily in man-made containers. Larvae ingest microbes routinely throughout development [5], and their gut microbiota is primarily composed of microbes orally acquired from the environment [6,7]. As adults, *Ae. aegypti* also host bacterial communities in their digestive tract. The sources of these bacteria are not clear. Bacteria can be transstadially transmitted from larvae to adults which may account for some of the bacteria found in adults [6,8,9], but it is also hypothesized that adults acquire bacteria when nectar feeding [10].

The microbial communities associated with adult *Ae. aegypti* mosquitoes have garnered much attention because of their sizeable impact on life history traits and susceptibility to infection by human pathogens (reviewed in [11]). For example, the microbiota of adults can impact susceptibility to dengue virus, blood meal digestion, and fecundity [12,13]. Additionally, experimentally introducing certain microbes to the digestive tract of adult female mosquitoes significantly impacts mosquito longevity and susceptibility to arboviruses [14–17].

The vast majority of investigations into the adult microbiota of mosquitoes have focused on the midgut portion of the female digestive tract. This is because the midgut is where the blood meal is stored during digestion and is the tissue through which blood borne pathogens invade [18,19]. However, microbes are also found in other compartments of the adult female mosquito digestive tract, such as the crop, a nectar storage organ anterior to the midgut [20]. In mosquitoes and other adult hematophagous Diptera (e.g., Ceratopogonidae [21]; Simuliidae [22]), the crop is a ventral diverticulum of the esophagus (foregut) separated by a muscular valve [23]. Sensory cells signal the esophageal valve to direct meals rich in sugar (e.g. nectar or artificial sugar supplements) to the crop and meals low in sugar (e.g. blood meals) to the midgut [24–26]. Sugar-rich meals are stored in the crop until they are pumped by peristaltic

contractions of the crop back into the digestive tract for digestion [27]. This allows the mosquito to retain sugars until they are needed for energy intensive activities, such as flight [28]. The crop is generally considered only a storage organ because its cuticular lining prevents most nutrients from being absorbed [23]. However, some carbohydrate digestion (e.g., sucrose catabolism) may occur here because some salivary enzymes are also retained with the meals in the crop [23,25]. The Dipteran crop is known to maintain stable populations of microbes (reviewed in [28,29]) which may also aid in the digestion of these sugars.

Little research to date has focused on the microbiota in the mosquito crop. A better understanding of the mosquito crop microbiota is of interest because (1) microbes in the crop could be contributing to the physiological impacts of the microbiota described above, either independently or in concert with the microbial communities in other regions of the gut, and (2) it is possible that microbes from the crop could influence formation of the midgut microbiota if they are passed from the crop to the midgut in nectar.

As far as we are aware, only one study has investigated the microbiota composition in the crop in *Ae. aegypti* [20]. Gusmão et al. dissected crop tissues from recently eclosed laboratory reared adult females that had not yet fed on sucrose or blood. They found that the crop was dominated by bacteria in the genera *Serratia* and *Bacillus* as well as yeasts in the genera *Pichia* and *Candida*. They also report acidification of the crop and the production of acid by a strain of *Serratia* isolated from this tissue. In laboratory reared *Aedes albopictus*, Guégan et al. [30] compared the microbiota of the crop with that of the midgut in the same individuals and found that alpha and beta diversity of bacterial and fungal communities did not significantly differ between the tissues. They reported that bacteria from the families Weeksellaceae and Burkholderiaceae were most abundant in both crops and midguts in female *Ae. albopictus* and that Corynebacteriaceae was more abundant in crops than in midguts.

In the present work, we used culture-dependent (culturing bacteria on growth media) and culture-independent (qPCR and high-throughput amplicon sequencing of the bacterial 16S rRNA gene) methods to quantify and profile the bacterial communities in paired crops and midguts from adult *Ae. aegypti* females.

## Methods

*Mosquito rearing*: *Aedes aegypti* Singapore (Sing) mosquitoes were established from larvae collected in the field in Singapore in 2010 [31]. For strain maintenance, we reared Sing strain larvae at 27˚C and 80% residual humidity on a 14:10 light:dark photocycle. We reared larvae in reverse osmosis (RO) water with ad libitum access to larval food (liver powder, tropical fish flake food, and rabbit food pellets mixed in a 2:1:1 ratio and autoclaved) and provided adults with ad libitum access to 10% sucrose. To rear mosquitoes for experiments, we hatched eggs in a vacuum in RO water. When rearing mosquitoes for the culture-dependent experiment, we rinsed eggs one time with 3% bleach, then twice with RO water before hatching. In all cases, we thinned larvae to approximately 250 larvae per pan and provided ad libitum access to larval food. We did not attempt to manipulate or control the microbial environment of the larvae and allowed the bacterial community to spontaneously form during larval development. Adults were maintained on 10% sucrose until dissection.

*Dissection and sample preparation*: We dissected adult females at 4–6 days post eclosion. Mosquitoes used for culture-independent and those used for culture-dependent experiments came from separate batches of mosquitoes. In both cases, we externally sterilized females with 70% EtOH for one minute, then rinsed twice with filter-sterilized 1X phosphate buffered saline (PBS). We dissected midguts and the ventral diverticula of the foreguts (i.e. crops) from adult mosquito bodies in sterile 1X PBS on glass slides (pre-sterilized with 70% EtOH) and moved

the tissues to a sterile pool of 1X PBS. Using sterile forceps, we separated the crop at the diverticular valve. The midgut was separated from the crop immediately anterior to the proventriculus and the proventriculus was included with the midgut. We rinsed both tissues twice in separate pools of sterile 1X PBS. Between each step of the dissection (e.g., after removing the digestive tract, after transferring the crop, after transferring the midgut), we cleaned the forceps with 70% EtOH.

## Culture-dependent experiments

*Sample processing*: We dissected midguts and crops from eight adult females as above and transferred each to separate 1.5ml microcentrifuge tubes containing 150μL 1X PBS without pooling (8 crop samples and 8 paired midgut samples). We kept all samples on ice until homogenizing them using a sterile pestle. We then serially diluted all samples $10^{-2}$ and $10^{-4}$ with additional sterile 1X PBS and spread 50μL of each dilution plus the undiluted sample on tryptic soy agar (TSA) and M9 minimal media using sterile beads. We chose TSA because it is a high-nutrient media commonly used to isolate diverse environmental bacteria. We also used a low-nutrient media (M9) because use of only a rich media can favor fast-growing species and prevent isolation of rare and/or slow-growing species [32–34]. Plates were incubated at room temperature for 72 hours and then transferred to 4˚C.

*Bacterial colony characterization and quantification*: After 72 hours of bacterial growth, we counted colony forming units (CFUs) and grouped them by colony type using morphological characteristics (e.g., size, color, border, elevation). Counts were made on the lowest dilution plate where CFUs could be readily identified (generally <100 CFU/plate) and back calculated to determine the total number of each colony type present within each tissue. We re-isolated each colony type on fresh media and sequenced the 16S rRNA gene by colony PCR using primers 27F and 1492R [35].

We manually trimmed sequences for each colony type to remove low quality sections and then aligned forward and reverse sequences using the BioEdit Sequence Alignment Editor ([36], v7.2.5). If sequences were too short to be aligned or we were only able to obtain high quality sequence in one direction, we used partial sequences. We then used NCBI Nucleotide BLAST [37] with the Megablast algorithm and default parameters to query the Nucleotide Collection (nr/nt) to identify each colony type. We assigned colony type identities using the result with the highest percent identity as long as both the percent identity and query cover were higher than 98%. In two cases, we limited our identification to only the family level (Enterobacteriaceae) because the results showed multiple genera with >98% identity and query cover. All colony types with sequences ≥ 97% similar were combined into a single operational taxonomic unit (OTU), otherwise they were assigned an arbitrary number to indicate different OTUs (e.g., Enterobacteriaceae 1). Abundances and sequences of each colony type can be found in S1 File.

## Culture-independent experiments

*Sample processing*: For culture-independent experiments, we dissected crops and midguts from eight individuals and pooled each tissue type. This was repeated three times (24 crops and 24 paired midguts, pooled to make 3 total pools of 8). In addition, we dissected crops and midguts from five individuals that were not pooled (5 crop samples and 5 paired midgut samples). We were uncertain how many bacteria are present in crops and thought it likely these tissues would have low biomass. To increase the likelihood of successful community profiling, we chose to analyze individuals and pools for high throughput sequencing. We also collected two contamination control buffer blanks that consisted of lysis buffer handled identically to an

experimental sample but without tissue added. We placed all samples in 200μL lysis solution from the ZymoBIOMICS DNA Miniprep Kit (Zymo Research, Irvine CA, USA) in 1.5mL microcentrifuge tubes on ice and froze them at -80˚C until DNA extraction. We extracted DNA using the ZymoBIOMICS DNA Miniprep Kit according to the manufacturer's instructions with the following adjustments: all samples were homogenized manually using sterile pestles treated with DNA Away (Thermo Fisher) and rinsed with sterile water, and we eluted DNA in 75μL filter-sterilized water.

*qPCR*: To assess the total amount of bacteria in each tissue sample, we conducted qPCR targeting the 16S rRNA gene as described in [38]. Briefly, each reaction contained 7.5 μL SYBR master mix (Applied Biosystems), 0.35 μL of each primer (primer starting concentrations were all 10 μM), 5 μL template (diluted 1:20), and MilliQ water to a final volume of 15 μL. qPCR conditions were as follows: 95˚C for 10 min, (95˚C for 15 s then 60˚C for 1 min) × 40 cycles. A melt curve was performed after all reactions to verify single product amplification. We estimated 16S copy number of each sample using a standard curve generated from serial dilutions of an *Escherichia coli* 16S PCR amplicon stock of known concentration. To amplify 16S from *E. coli*, we used the same 16S qPCR primers as for the experimental samples. Universal primers used for 16S qPCR are described in [39].

*High-throughput 16S amplicon sequencing and data processing*: High-throughput 16S amplicon sequencing was performed as described in [38] at the Microbial Analysis, Resources, and Services center at the University of Connecticut. Briefly, 5ng of DNA was used as template in a PCR with 515F and 806R primers with Illumina adapters and eight basepair dual indices [40,41]. Many samples (S1 File) did not amplify using the standard 30 cycles, and in these cases, an additional five cycles were performed which resulted in successful amplification. The contamination control buffer blanks failed to amplify but were still included in the sequencing reaction to account for any potential contamination. 250bp paired-end sequencing was performed on a MiSeq system using the MiSeq Reagent Kit V2 (Illumina, Inc.).

Sequences were demultiplexed using onboard bcl2fastq. We completed all bioinformatic analysis using R software v4.1 [42]. We used the *DADA2* R package v1.16 [43] with default parameters specified by the author ([44]; https://benjjneb.github.io/dada2/tutorial.html) to identify amplicon sequence variants (ASVs), assign each ASV a taxonomic identity using the RDP classifier with training set v18 [45], and generate count data for each ASV. The specific code we used for sequence processing, downstream bioinformatics, and statistical analyses can be found in S2 File. We imported the data to the *phyloseq* R package v1.30.0 [46] for downstream bacterial community analyses and visualization. We created a phyloseq object from the ASVs identified by DADA2. We then removed ASVs not from the Bacteria kingdom and those classified as either mitochondria or chloroplast. As low biomass samples are prone to the effect of environmental and reagent contamination [47,48], two buffer blank controls were included in the experimental design. Before any other filtering step was applied to the phyloseq object, we explored the composition of these controls and their similarities to the experimental samples to better understand the nature of potential contamination and address it. The most abundant taxon in the buffer blank controls was *Acinetobacter*, and *Elizabethkingia* was present as well. These taxa are associated with *Aedes* mosquitoes [49] and their natural environment [10,50,51] but are also known laboratory contaminants and are commonly found in this environment [48,52]. The average number of reads in the controls accounted for a very low percentage (0.51%) of the mean reads per sample in our dataset (S1 Fig), and a principal coordinate analysis (PCoA) revealed that buffer blank controls clustered separately from experimental samples (S2 Fig). Therefore, we proceeded with the analyses despite this minor contamination. As an additional quality control step, we used the *decontam* R package v1.6 [53] with default prevalence filtering parameters to remove ASVs identified as contamination.

Buffer blank control samples were then removed from the dataset. Finally, we removed ASVs accounting for fewer than 5 total reads and then scaled reads per sample based on the smallest read count as suggested by [54] in order to preserve proportions.

## Statistical analysis

For both the culture-dependent experiments and the qPCR of the culture-independent experiments, we assessed the differences in microbiota size between the crop and midgut tissues. We used either a t-test or Kruskal-Wallis test in R (v4.1, [42]) depending on whether the data conformed to the assumptions of a parametric test. We also assessed whether the variance of the microbiota in each tissue was significantly different using Levene's test in the *car* package v3.0–13 [55].

*Microbial ecology analyses*: Unless stated, we used R (v3.3.2) and the Rstudio environment [56] to perform all the statistical analysis and graphical representations for the high-throughput amplicon sequencing analysis. To measure alpha diversity, we calculated the observed richness and the Gini-Simpson index (1-D) [57] for each sample with *phyloseq*. Differences between tissues were evaluated by a Kruskal-Wallis test separately for each sample type (single and pooled) using *ggpubr* v0.4.0 [58]. We assessed beta diversity and clustering profiles using non-constrained (PCoA) and constrained ordinations (Canonical Correspondence Analysis, CCA) based on Bray-Curtis dissimilarities. The PCoA plot was generated with the *microeco* package v0.3.3 [59] and the CCA plot with the *vegan* package v2.5.7 [60]. To partition the variance within the data matrix, we used the adonis2 function in *vegan* to run a permutational multivariate analysis of variance (PERMANOVA) to test the effects of tissue (crop or midgut), sample type (single or pooled), and the most abundant taxa in the dataset on variation in bacterial community composition. We then applied two approaches to identify microbial community members significantly associated with each tissue type. We chose a combination approach as recommended by [61] to prevent misidentifying discriminant features due to technical and analytical artifacts (e.g. sequencing depth and filtering of rare ASVs). The first approach, LEfSE [linear discriminatory analysis (LDA) effect size] [62], identifies biomarkers that are differentially abundant between treatments (tissue type in our case) and ranks them based on their association with each treatment level. The analysis was run with an LDA cutoff value of 3.0 and p-value of 0.05 using the *microeco* package. The second approach, indicator species analysis, relies on community ecology principles to identify taxa reflective of their niche state and predicts the diversity of other community members. We searched for indicator ASVs using the multipatt function within the *indicspecies* package v1.7.0 [63]. To perform functional profile predictions of the prokaryotic communities within each tissue, we used FAPROTAX within the *microeco* package [64]. This algorithm maps the ASV compositional profile of each sample against a curated database of metabolic and ecologically relevant functions based on literature associated with cultured bacterial representatives. We tested the significance of the functional profile differences between tissue groups with a Wilcoxon rank-sum test using the *ggpubr* package reporting p values < 0.05 after a false discovery rate correction.

## Results

### Size of the microbiota is more variable in midguts compared to crops

We first estimated the number of bacteria in each sample by quantifying 16S rRNA gene copy number using qPCR. Pooled samples showed a different pattern than single (non-pooled) samples. The variance was highly similar between tissues for pooled samples (Levene's test for equal variances: F = 0.256, p = 0.640) but in single samples, the variance of midguts was significantly larger than crops (Levene's test for equal variances: F = 6.12; p = 0.038). Pooled midgut

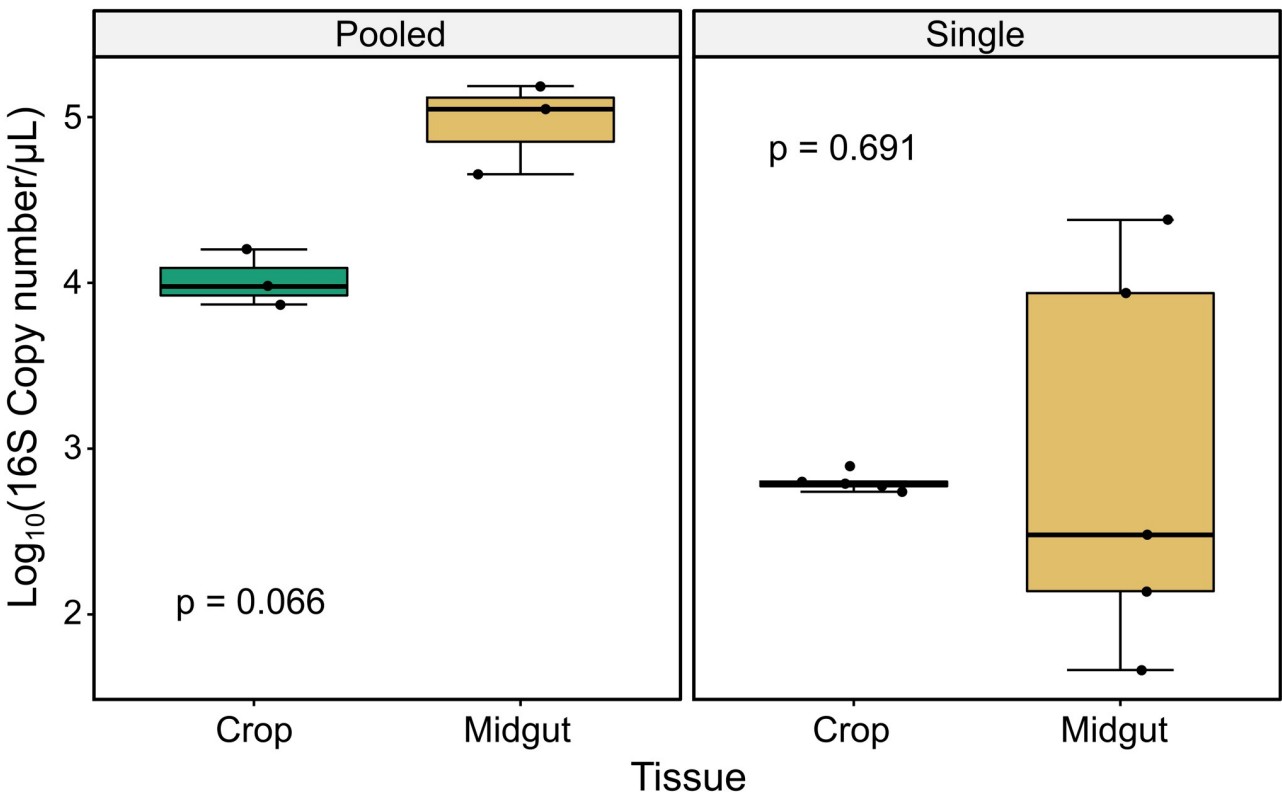

**Fig 1. Microbiota size measured using qPCR is more variable in midguts compared to crops.** We measured copy number of the 16S bacterial rRNA gene using qPCR. Samples were either pooled (left panel) in groups of 8 (n = 3 pools per tissue type) or assayed as individual single tissues (right panel) (n = 5 samples per tissue type). For pooled samples, variance was similar between tissues (Levene's test for equal variances: F = 0.256, p = 0.640), but for single samples the variance among midgut samples was significantly larger than crop samples (Levene's test for equal variance, F = 6.12, p = 0.038). For pooled samples, median $\log_{10}$(16S copy number/µl) trended higher in midguts (median = 5.05) than in crop samples (median = 3.98) though the difference between tissues was not significant using a paired t-test (t = 3.69, p = 0.066). For single samples, median values were similar for midguts (median = 2.48) and crops (median = 2.79) and were not significantly different using a Wilcoxon rank sum test (p = 0.691).

samples trended toward a higher $\log_{10}$ 16S copy number (median$_{\text{midgut\_p}}$ = 5.05) than pooled crop samples (median$_{\text{crop\_p}}$ = 3.98), though the tissues were not significantly different using a paired t-test (t = 3.69, p = 0.066, Fig 1). The median copy number for single midgut samples was highly similar to that of single crop samples (median$_{\text{midgut\_s}}$ = 2.48; median$_{\text{crop\_s}}$ = 2.79), and tissues were not significantly different using a Wilcoxon rank sum test (p = 0.691, Fig 1).

We also estimated the total number of cultivable bacteria in each tissue by homogenizing single, paired crops and midguts from adult female *Ae. aegypti* and culturing the homogenate on TSA and M9 media at multiple dilutions. Variances were significantly different between tissues for both media types as measured by Levene's test (TSA: F = 5.521, p = 0.034; M9: F = 11.554, p = 0.004). The median number of CFU/sample trended higher in midguts (median$_{\text{midgut\_TSA}}$ = $4.0 \times 10^{6}$, median$_{\text{midgut\_M9}}$ = $6.5 \times 10^{6}$) than in crops (median$_{\text{crop\_TSA}}$ = $1.9 \times 10^{3}$, median$_{\text{crop\_M9}}$ = $2.9 \times 10^{3}$, Fig 2) for both media types, but these differences were not statistically significant as measured by a Kruskal-Wallis test ($p_{\text{TSA}}$ = 0.074, $p_{\text{M9}}$ = 0.093).

## Community composition of crop and midgut tissues using culture-independent and culture-dependent approaches

We sequenced the bacterial community of 16 samples (3 pools of 8 crops, 3 pools of 8 midguts, 5 individual crops, and 5 individual midguts) and 2 buffer blank controls resulting in 534462

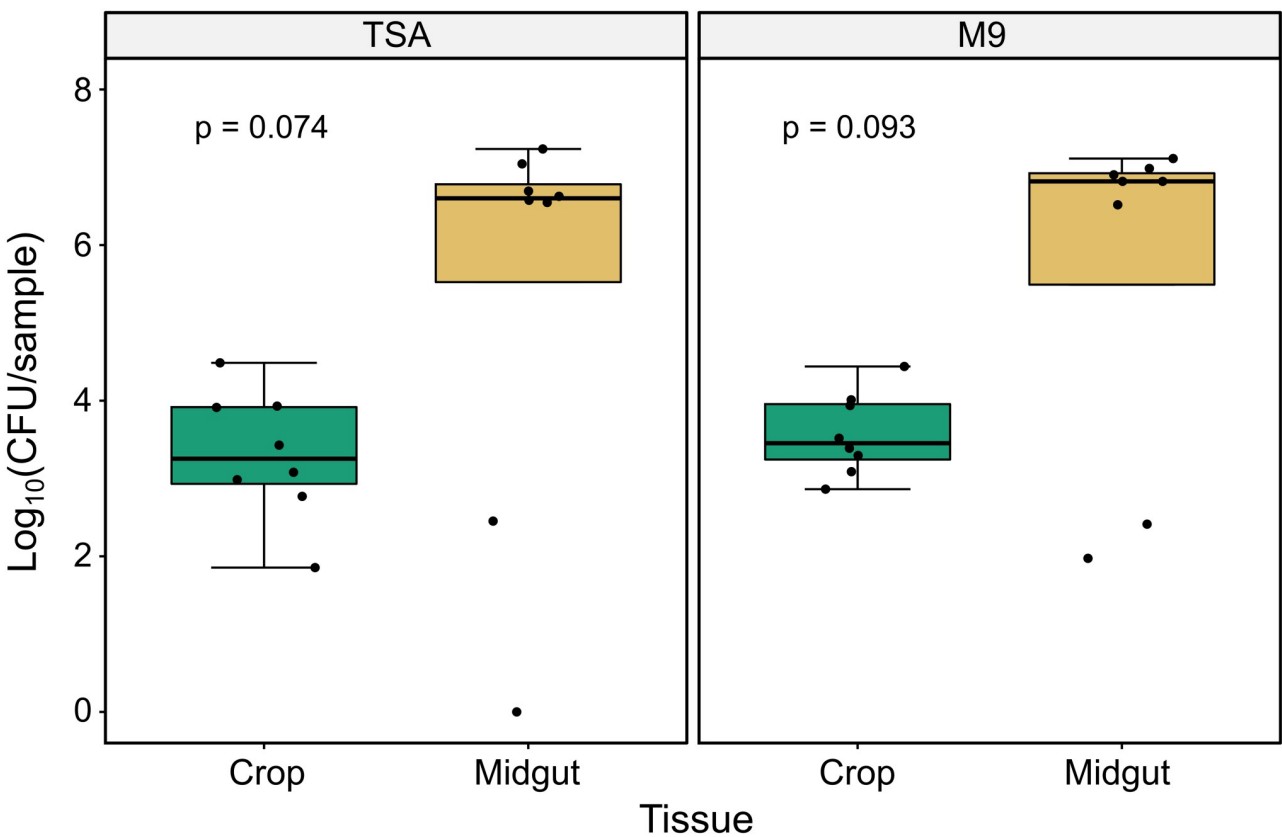

**Fig 2. Bacterial load in crops and midguts of adult female *Ae. aegypti*.** Individual crops and midguts from eight adult females were homogenized and cultured on TSA (left panel) and M9 (right panel) media. Variance was significantly different between tissues for both media types (TSA: F = 5.521, p = 0.034; M9: F = 11.554, p = 0.004) as measured by Levene's test for equal variances. Tissue type was not a significant predictor of bacterial load on either media (TSA: p = 0.074; M9: p = 0.093) as measured by Kruskal-Wallis test.

amplicon reads; all but two samples reached saturation (S3 Fig). One individual midgut sample failed during sequencing and no data were obtained for that sample. After buffer blank removal, decontamination, and quality filtering, the dataset was comprised of 531713 reads, with samples ranging between 9246 to 65275 reads, a median of 37380 reads, and 0.80 sparsity. After scaling to address the effect of sequencing depth, we retained 9221 reads as the minimum read number per sample. In total, the dataset comprised 134 ASVs taxonomically distributed into 10 bacterial phyla. From them, 97% were classified within 6 phyla as follows: Proteobacteria (48.8%), Firmicutes (23.7%), Actinobacteria (11.1%), Chloroflexi (6.6%), Planctomycetes (3.7%), and Bacteroidetes (2.9%).

All crop samples (pooled and single) were dominated by bacteria from the family Acetobacteraceae. Bacteria from the family Weeksellaceae were also present in appreciable amounts in many crop samples but never reached higher than 9.8% of the total community (Fig 3). Midgut samples were dominated by Weeksellaceae when pooled, but single samples were more variable. Two of four single midguts had near total dominance of Weeksellaceae while the other two midguts contained Acetobacteraceae and multiple other families (Fig 3).

We also evaluated community composition by culturing 8 single, paired crops and midguts on TSA and M9 media (Fig 4). These were different individuals than those used in Fig 3. We characterized and counted bacterial CFUs based on morphological characters, then sequenced the 16S rRNA gene from a representative isolate to identify each colony type to genus. Samples

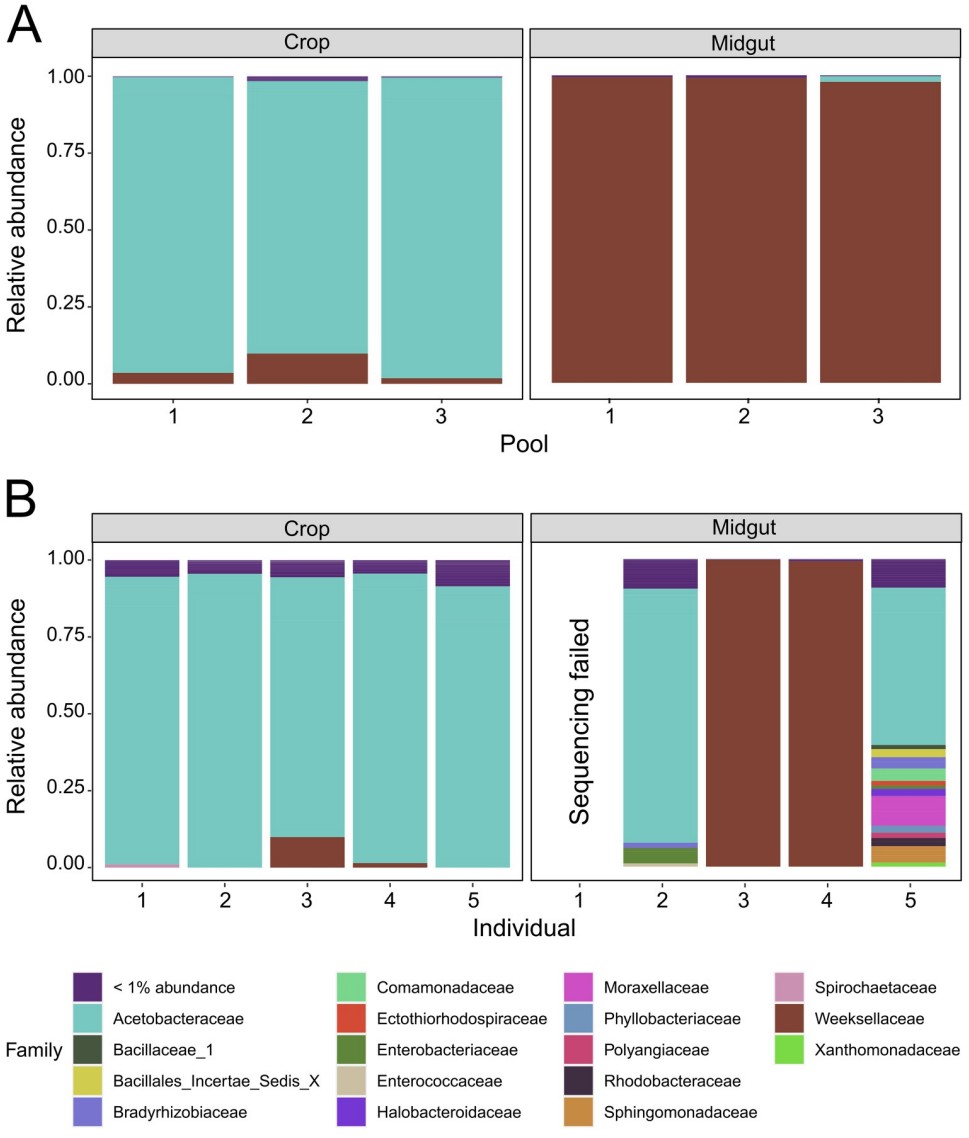

**Fig 3. Composition of microbial communities in crops and midguts of *Ae. aegypti* adult females using a culture-independent approach.** Relative abundances of bacterial families are clustered by tissue (labeled across the top) and sample type. In (A), samples are pooled in three groups of eight. Tissues from each numbered pool are paired, i.e., they were collected from the same eight individuals. In (B), samples are single (un-pooled) paired tissues, i.e. each numbered crop came from the same individual as the matching numbered midgut. Bacterial communities in crops are dominated by Acetobacteraceae. Midguts are mostly dominated by Weeksellaceae, though individual midgut samples 2 and 5 lack Weeksellaceae and are dominated by Acetobacteraceae.

cultured on both types of media showed similar, but not identical, results. The largest difference between media types was that we did not observe *Tanticharoenia* bacteria on TSA media (Fig 4A) but did using M9 media (Fig 4B). Similar to what was observed in the culture-independent experiments, genera in the family Acetobacteraceae (i.e., *Asaia* and *Tanticharoenia*) dominated the crop samples (Fig 4A and 4B). *Elizabethkingia* (Family Weeksellaceae) were present in 75% of midgut samples and represented more than 99.9% of observed CFUs when present (Fig 4A and 4B). *Elizabethkingia* were also dominant in 37.5% of crops and present in

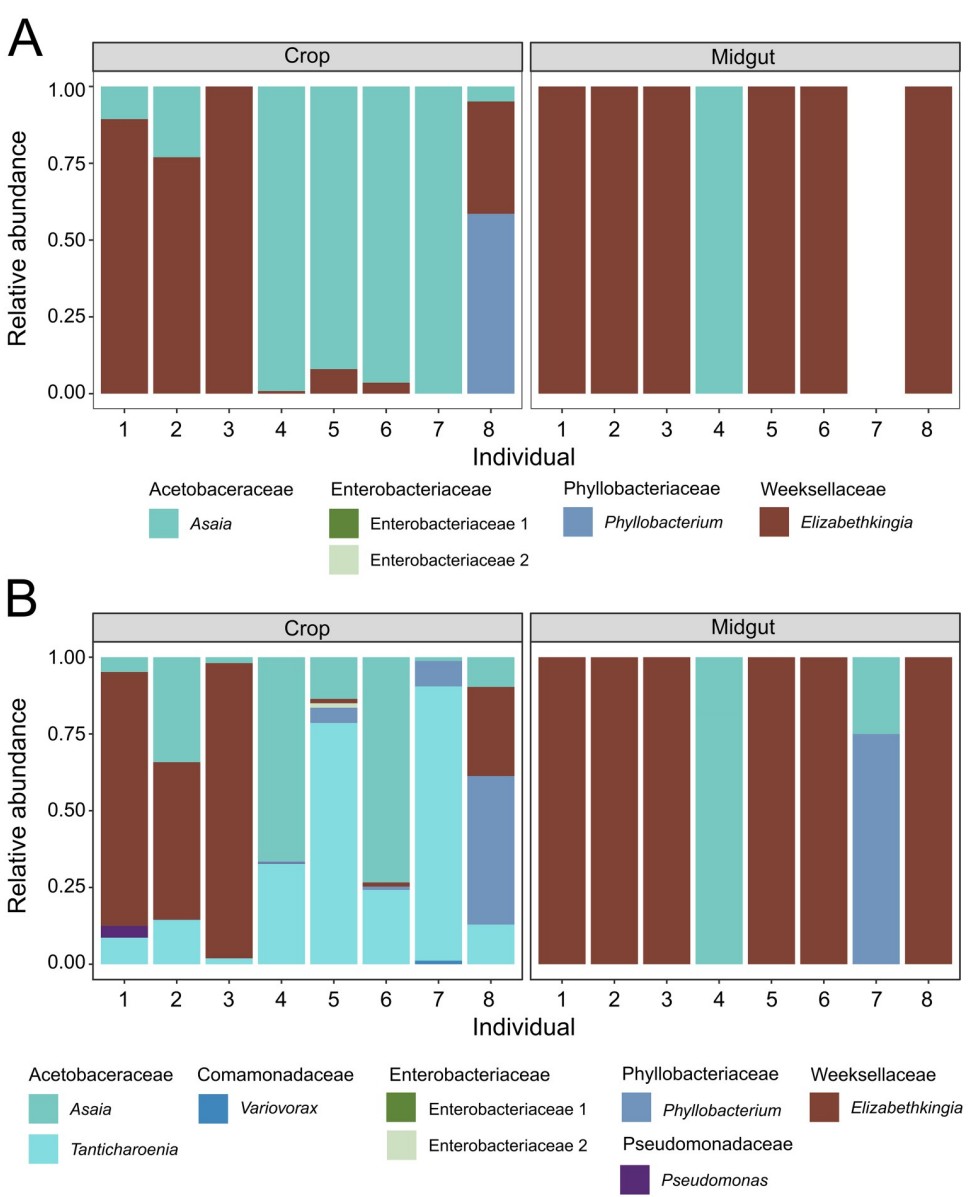

**Fig 4. Composition of microbial communities in crops and midguts of *Ae. aegypti* adult females using a culture-dependent approach.** Relative abundances of bacterial genera are clustered by tissue (labeled across the top). Individual crops and midguts from eight adult females were homogenized and an aliquot from each was then cultured on TSA (A) and M9 (B) media. In all cases, samples are single (un-pooled) paired tissues, i.e., each numbered crop came from the same individual as the matching numbered midgut. Bacterial communities in crops are dominated by *Asaia*, *Tanticharoenia*, and *Elizabethkingia*. Midguts are almost all dominated by *Elizabethkingia*, though individual midgut samples 4 and 7 lack this genus.

varying amounts in all but one. However, *Elizabethkingia* made up a smaller proportion of CFUs in crops compared to midguts (Fig 4A and 4B).

## Bacterial community diversity analysis in culture-independent experiments

Following our initial observations of microbial community composition, we next analyzed the high-throughput amplicon sequencing data to evaluate whether alpha and beta diversity differ

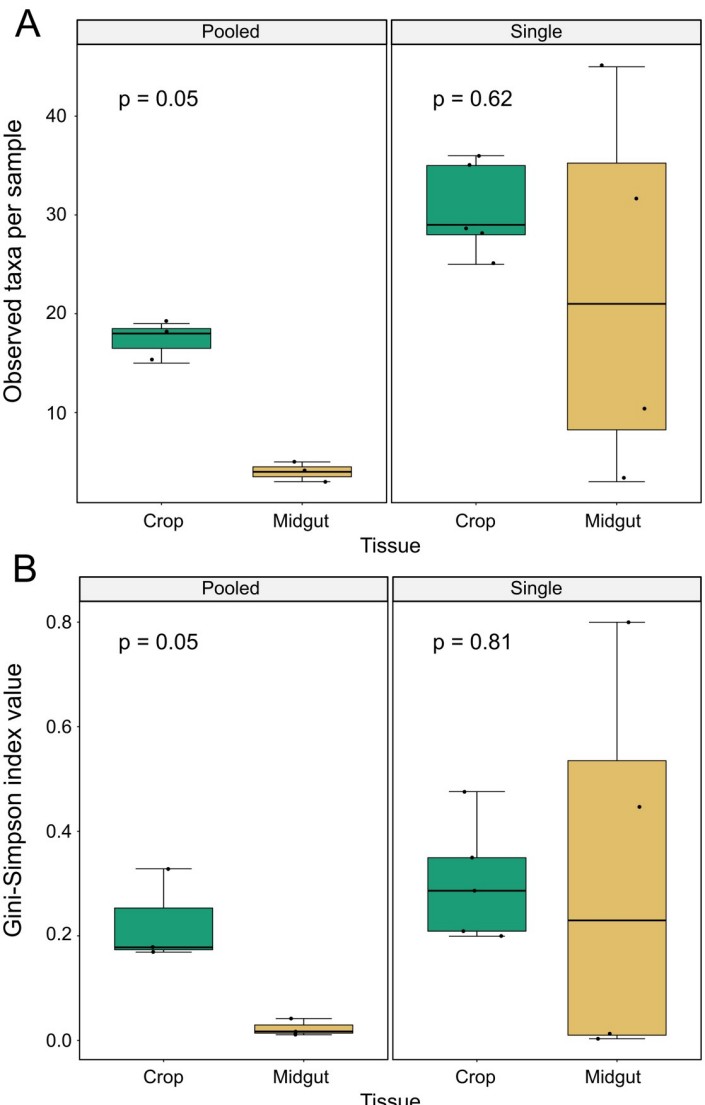

**Fig 5. Alpha diversity differs between crop and midgut samples.** We measured alpha diversity as observed ASVs (A), and the Gini-Simpson index (B). For pooled samples (left panels), both measures of alpha diversity were marginally significantly higher in crops than midguts (Kruskal-Wallis test, p = 0.05 for both measures). Variance among crops was significantly larger than midguts for Gini-Simpson indices (Levene's test: F = 10.08, p = 0.03) but not for observed taxa values (Levene's test: F = 2.28, p = 0.205). For single samples (right panels), tissues were not significantly different for either measure, but midgut samples showed a much larger range of values than crops for both observed taxa values (Levene's test: F = 18.70, p = 0.003) and Gini-Simpson indices (Levene's test: F = 9.92, p = 0.001).

by tissue type and whether any specific bacterial taxa are significantly associated with crops and/or midguts.

*Bacterial communities within pooled crop samples are more diverse than midguts*: For pooled samples, alpha diversity measured as observed ASV number (Fig 5A, p = 0.05) and Gini-Simpson index (Fig 5B, p = 0.05) was marginally higher in crops than midguts when evaluated by a Kruskal-Wallis test. Using Levene's test for equal variances, we determined that the variance of pooled samples was similar between tissues for observed ASV number (F = 2.28, p = 0.205) but higher in crops than midguts for Gini-Simpson (F = 10.08, p = 0.03). For single samples, alpha

diversity was not significantly different between tissues, but Levene's test showed much higher variance among midgut samples than crops for observed ASV number (F = 18.70, p = 0.003) and Gini-Simpson (F = 9.92, p = 0.001).

*Composition and structure of microbiota in crops significantly differs from that in midguts*: We executed a PCoA on the high-throughput amplicon sequencing data to explore how tissue (crop and midgut) and sample type (pooled or single) drove the community profile variation among samples (Fig 6A). The first two components encompass 98.2% of the total variance (91.7% PCoA1 and 6.5% PCoA2). Samples predominantly grouped in homogeneous clusters by tissue type, separating themselves along axis 1. Two midgut samples digressed from this diversity profile and clustered separately. These two data points represent single midguts samples 2 and 5 (Fig 3) which had low numbers of Weeksellaceae and a larger number of Acetobacteraceae. In order to test the significance of tissue and sample type as driving variables of beta diversity, we performed a PERMANOVA. This analysis revealed tissue as the only significant variable ($R^2$ = 54.81 p = 0.003). Sample type accounted for a non-significant 6.6% of variance (p = 0.107), and no significant interaction among the effector variables was detected. The communities were dominated by reads from multiple ASVs that map to the genera *Elizabethkingia* (Family: Weeksellaceae), *Tanticharoenia* (Family: Acetobacteraceae), and/or *Asaia* (Family: Acetobacteraceae) (Fig 3). Therefore, we conducted a CCA (Fig 6B) to address how the three most abundant ASVs in the dataset (which map to *Tanticharoenia*, *Elizabethkingia*, and *Asaia*) may be driving the diversity profiles detected. *Elizabethkingia* reached maximum abundance in midgut samples, whereas both *Tanticharoenia* and *Asaia* were maximally abundant in crops. Tight clustering of samples suggests that tissue and dominant taxa included in the analysis account for a large amount of variance. A second PERMANOVA considering abundance counts of *Elizabethkingia*, *Tanticharoenia*, and *Asaia*, and their interactions with tissue type as variables (Table 1) confirmed that the variance is significantly driven by these dominant taxa and their respective interactions with tissue type, accounting for 99.68% of the total beta diversity.

*Elizabethkingia and Tanticharoenia are biomarkers of midgut and crop tissues, respectively*: We used two methods to test for differentially abundant taxa with significant associations to each tissue. Linear discriminant effect size values estimated with the LEfSe tool (Fig 7) detected ASVs within the Bacteroidetes clade as consistently more abundant in the midgut. This was driven by ASV_2 and ASV_281, which were classified as g_*Elizabethkingia* (f_Weeksellaceae) and an unidentified member of f_Weeksellaceae, respectively, (LDA = 3.48, p < 0.01). LEfSe also detected ASVs within the Proteobacteria clade as biomarkers for the crop. Within this clade, ASV_1 (the most abundant amplicon in the dataset), classified as g_*Tanticharoenia* (f_Acetobacteraceae), had an LDA of 3.49 (p < 0.01). Although below the LDA score cutoff, ASV_3, identified as g_*Asaia* (f_Acetobacteraceae), was also detected as a significant crop biomarker (LDA = 2.33, p < 0.01). Indicator species analysis detected two ASVs as reliable predictors for the crop. One was ASV_1 (g_*Tanticharoenia*) and the second was a low-abundance ASV that mapped to g_*Rubrobacter* (Table 2). No significant indicator ASVs were identified for the midgut using this method (Table 2).

*Functional prediction analysis reveals potential differences in microbial community metabolic function between crop and midgut*: FAPROTAX prospection, a tool used for metabolic phenotype predictions, revealed 34 functional categories represented in the microbial communities in the full dataset (Fig 8). Among them, 15 were shared between both tissues: aerobic chemoheterotrophy, anaerobic chemoheterotrphy, animal parasites or symbionts, aromatic compound degradation, chemoheterotrophy, fermentation, human associated, human gut, human pathogens all, mammal gut, nitrate reduction, nitrate respiration, nitrogen fixation, nitrogen respiration, and ureolysis. Fourteen functional categories were only found in crops: aliphatic

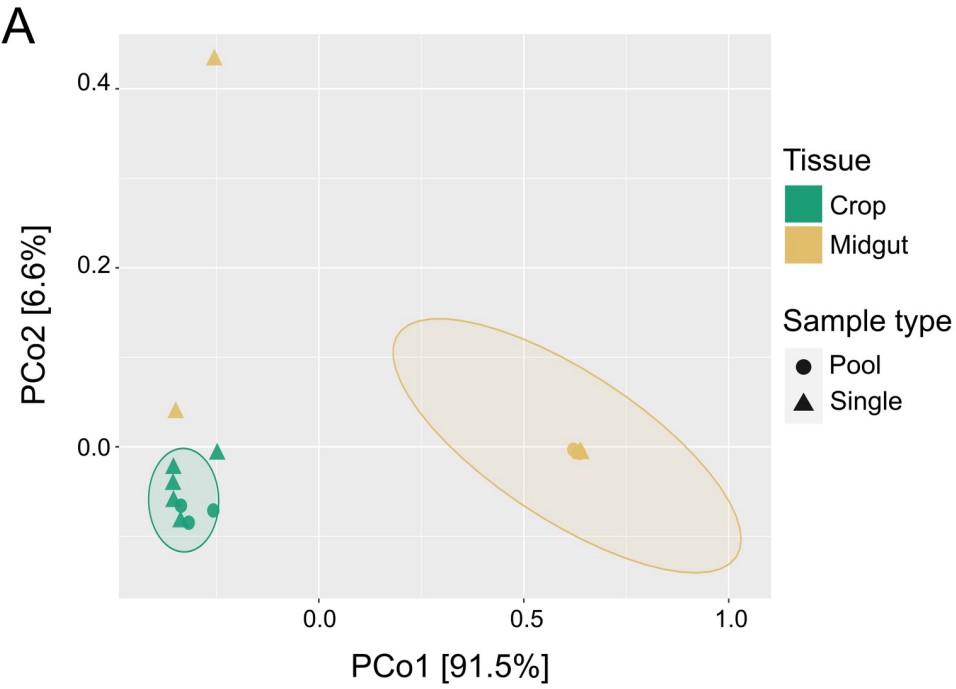

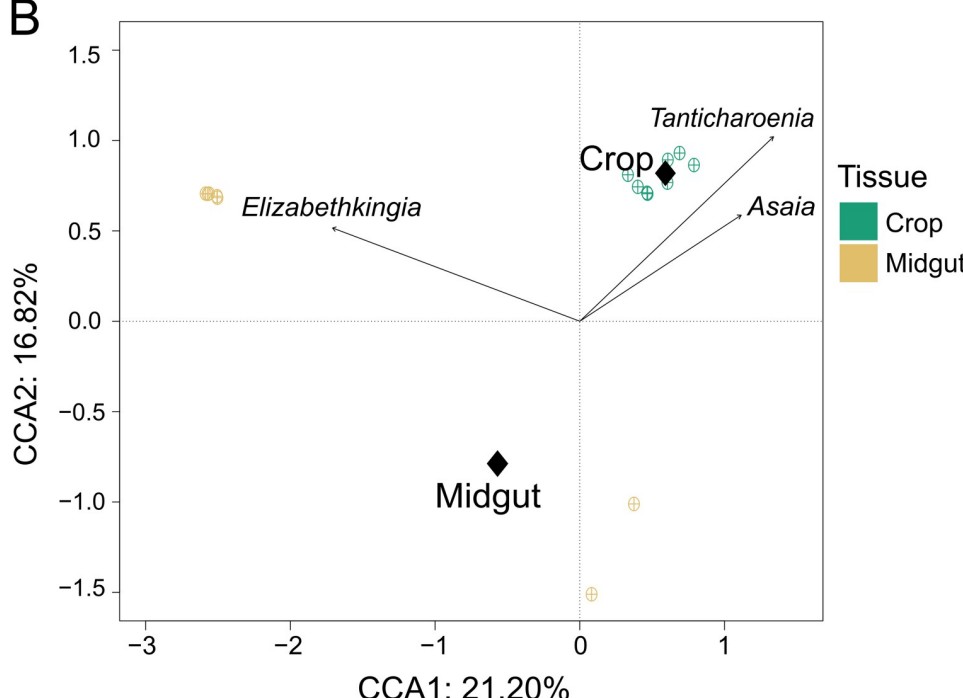

**Fig 6. Ordination analyses reveal significantly different bacterial community structures between crop and midgut tissues.** Beta diversity was estimated using Bray-Curtis dissimilarities. The unconstrained analysis (principal coordinates analysis, PCoA) (A) shows the samples cluster by tissue type predominately along axis 1 (91.7%). The constrained analysis (canonical correspondence analysis, CCA) (B) shows that the explanatory variables (Table 1) capture 21.2% and 16.82% of variation in CCA1 and CCA2, respectively. The group centroids of each tissue (represented by a black diamond) are located in opposing coordinates, indicating a difference between the communities. The abundance of the dominant taxa (*Asaia*, *Tanticharoenia*, and *Elizabethkingia*), shown as vector arrows, indicate that *Asaia* and *Tanticharoenia* are maximally abundant in crops while *Elizabethkingia* is maximally abundant in midguts.

**Table 1. A PERMANOVA to test for the association between beta diversity (quantified by Bray-Curtis dissimilarities) and explanatory variables representing tissue type and the raw abundance of the three dominant taxa in the data set.**

| Explanatory variable | $R^2$ | p-value |
|---|---|---|
| Tissue | 0.0291 | 0.001 |
| *Asaia* | 0.0281 | 0.001 |
| *Elizabethkingia* | 0.7446 | 0.001 |
| *Tanticharoenia* | 0.0439 | 0.001 |
| Tissue:*Asaia* | 0.0959 | 0.001 |
| Tissue:*Elizabethkingia* | 0.0048 | 0.001 |
| Tissue:*Tanticharoenia* | 0.0501 | 0.001 |
| Residuals | 0.0104 | |
| Total | 1.00000 | |

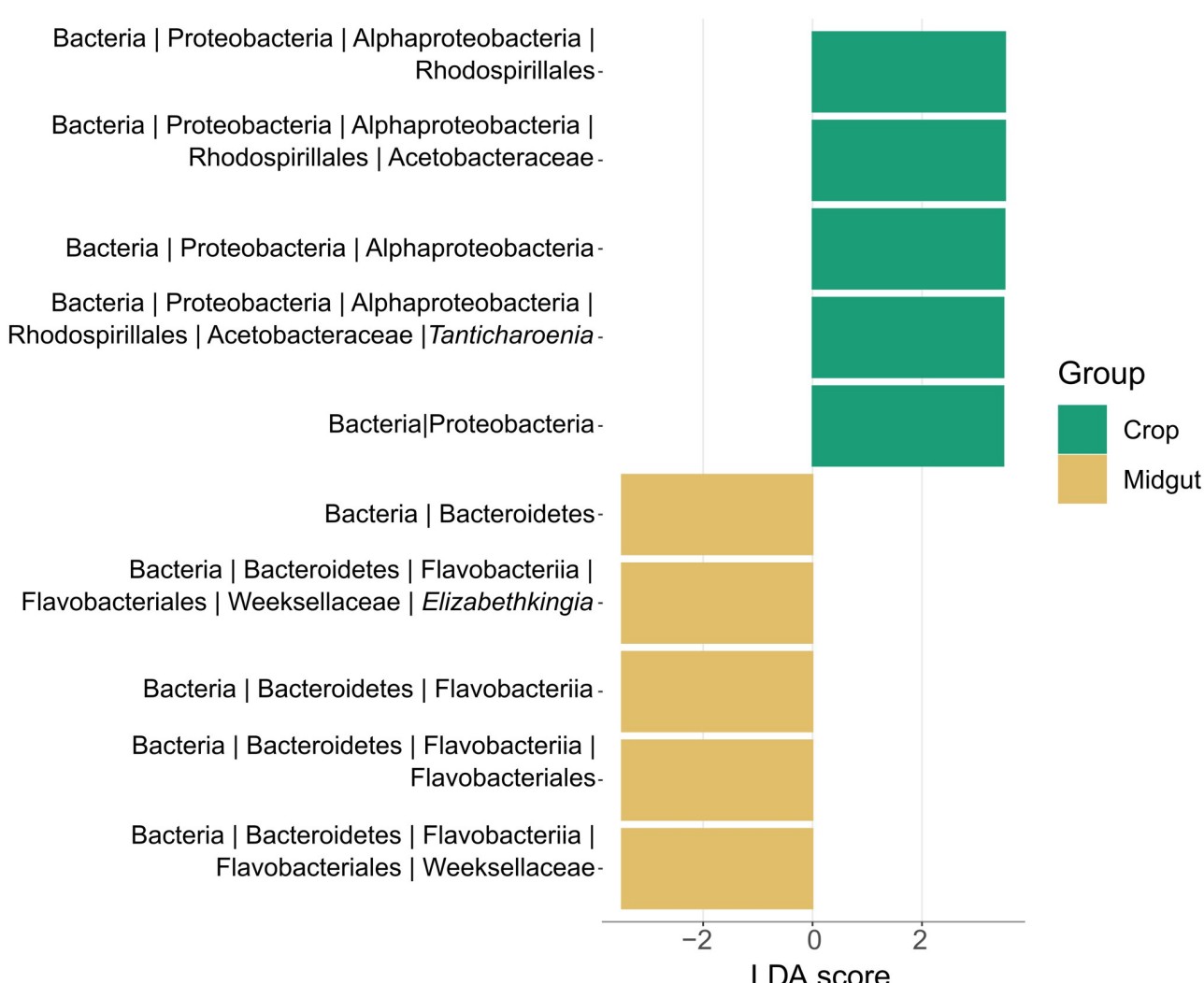

**Fig 7. Tissue specific biomarkers determined by LDA scores using LEfSe method.** At an LDA cutoff point of 3 (p < 0.01), two bacterial clades representing three ASVs were identified as significantly differentially abundant between tissues.

**Table 2. ASVs significantly associated with tissue samples as revealed by indicator species analysis.**

| Tissue | Taxa | Indicator statistic/Fidelity | p–value |
|---|---|---|---|
| Crop | *Tanticharoenia* | 0.934 | 0.001 |
| | *Rubrobacter* | 0.854 | 0.034 |
| Midgut | NONE | | |

non methane hydrocarbon degradation, chitinolysis, dark hydrogen oxidation, denitrification, human pathogens pneumonia, hydrocarbon degradation, knallgas bacteria, methanol oxidation, methanotrophy, methylotrophy, nitrate denitrification, nitrite denitrification, nitrite respiration, and nitrous oxide denitrification. Five functional categories were only found in midguts: anoxygenic photoautotrophy, anoxygenic photoautotrophy S oxidizing, manganese oxidation, photoautrotophy, and phototrophy. Of the 19 categories found in only one tissue, there were 11 singletons (inferred from only one sample). Chemoheterotrophy and aerobic chemoheterotrophy were ubiquitous among all samples (Fig 8). Chemoheterotrophy was the most abundant functional category in both tissues, accounting for 64.38 ± 3.00% in the crop and 48.96 ± 5.00% in the midgut. Comparative analysis revealed two functional categories that

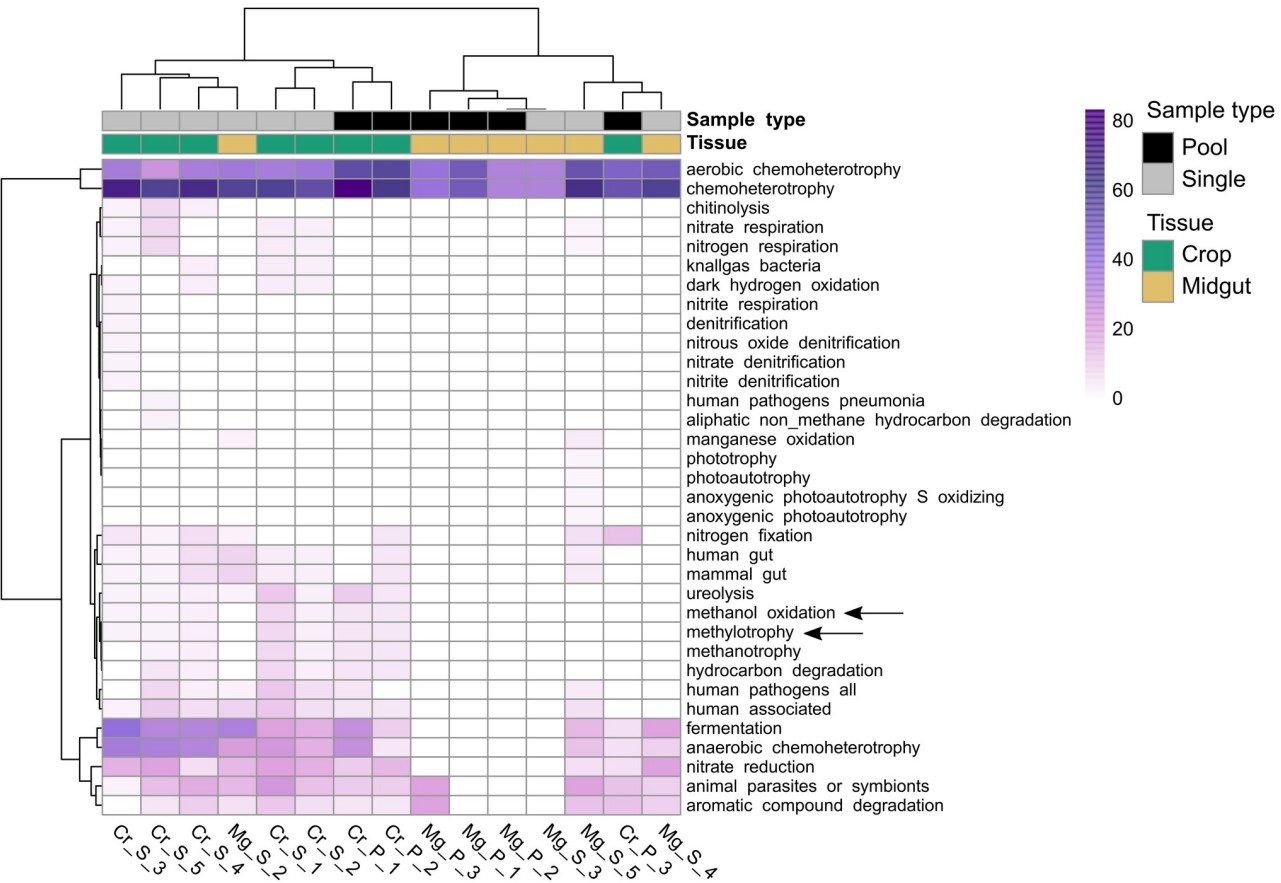

**Fig 8. Predicted community metabolic profiles reveal a predominant clustering pattern driven by tissue.** The heatmap represents the relative abundance of the predicted pathways (rows) in the community from each sample (columns) based on the FAPROTAX database. The color key highlights the tissue of origin (teal/beige) and sample type (black/gray) of each bacterial community. Statistical significance indicated by an arrow (p < 0.05) was tested by Wilcoxon rank sum test followed by an FDR correction.

were significantly more represented in crops than midguts: methylotrophy, which is binned within the carbon metabolism related functions in the FAPROTAX database, and methanol oxidation, which is binned in the broader "other annotated categories" in the database (Fig 8). Though not significant, fermentation, aromatic compound degradation, nitrate reduction, and animal parasites/symbionts were more commonly observed in crop samples, and at higher community proportions than in midguts (Fig 8).

## Discussion

In the current work, we used culture-dependent and culture-independent techniques to investigate the bacterial communities found in the crop and midgut of laboratory reared adult female *Ae. aegypti* mosquitoes. Our results reveal similarities and differences between the microbial communities of these two tissues.

The average size of the microbiota was not significantly different between tissues, though we did observe a trend toward crops having smaller and more diverse bacterial communities than midguts. However, microbiota size in midguts was dependent on whether the sample was pooled or single and by the composition of the microbiota. When measured singly, the size of the microbiota in midgut samples was revealed to be highly variable, whereas crop microbiota size was highly consistent between replicate samples in all cases. Moreover, midgut samples with the largest microbial communities were, in all cases, dominated by *Elizabethkingia* (Family: Weeksellaceae), while those with smaller microbial communities (i.e. the lower two outliers observed in the midgut samples in Fig 2) lacked *Elizabethkingia* (i.e. samples 4 and 7 from Fig 4), suggesting the presence or absence of this bacterial genus is a primary determinant of microbiota size in midguts. In contrast, crop samples had highly similar microbial community sizes regardless of composition.

We found that communities in both tissues were relatively simple, consisting of fewer than 20 ASVs. Our richness and within-sample diversity profiles are consistent with the low diversity reported in other studies that have sequenced the midgut and/or crop microbiota of laboratory reared sugar fed *Aedes* mosquitoes (e.g. [6,30,65–68]). Diversity within samples (alpha diversity) was higher in crops than midguts, but again, this was only for pooled samples; when sampled singly, we found that alpha diversity was more variable among midgut samples than among crops, and this was largely driven by the overdominance of *Elizabethkingia* in midguts. This suggests that analyzing single rather than pooled samples can be critical to assess bacterial community variation. Microbial communities differed significantly in composition and structure between tissues (beta diversity), however whether the samples were single or pooled was not a significant predictor of beta diversity. Crop tissues were primarily dominated by bacteria from the family Acetobacteraceae, while midguts were primarily dominated by bacteria from the family Weeksellaceae. The family Weeksellaceae was present in crops but at lower relative abundance than in midguts. Midguts also contained Acetobacteraceae but only in a minority of samples and only when Weeksellaceae was absent. A PCoA confirmed clustering of samples by tissue. A CCA coupled with a PERMANOVA showed that Acetobacteraceae and Weeksellaceae were significant drivers of differences between tissues and that Acetobacteraceae was significantly associated with crop tissues while Weeksellaceae was significantly associated with midguts. A biomarker analysis was consistent with these findings and identified *Tanticharoenia* and *Elizabethkingia* as markers of crops and midguts, respectively. We note that the differential abundance analysis methods differ in sensitivity and specificity, and LEfSE, which is a common method among vector biology microbiome studies, is susceptible to rare feature counts [69] and can result in false positives. Nearing et al. [61] stresses that some methods are reported interchangeably but can considerably differ in power and susceptibility to

experimental design. We therefore complemented our LEfSE analysis with an indicator species analysis, which did not give identical results but did corroborate the finding of *Tanticharoenia* as a biomarker for the crop.

Bacteria in the family Acetobacteraceae, specifically *Tanticharoenia* and *Asaia* in our samples, are considered "acetic acid bacteria" (AAB, [70,71]). *Tanticharoenia* was previously denominated as *Asaia* but was recently determined to be a distinct genus [71]. This group of bacteria is characterized by the common ability to catabolize sugars and produce acidic compounds from them (e.g., acetic acid). Specifically, *Tanticharoenia* produces acid from ethanol and multiple sugars, and can grow in the presence of 0.35% acetic acid [72]. Interestingly, *Asaia* is the only Acetobacteraceae that does not produce acetic acid from ethanol but can oxidize acetate and lactate to carbon dioxide [73].

Our midgut samples were highly dominated by *Elizabethkingia* (Family: Weeksellaceae). *Elizabethkingia*, and other bacteria from the phylum Bacteriodetes, are commonly reported in mosquito midguts, especially in laboratory settings but also in wild-caught adult specimens and *Ae. aegypti* larval development sites (e.g. [51,65,74–77]). *Elizabethkingia* has been previously shown to outcompete other bacteria in the midgut [78] and has been observed in midgut tissue of *Ae. aegypti* Singapore strain in high numbers [79]. Therefore, it was not unexpected that this genus was prevalent in our midgut samples; however, we consider it notable that *Elizabethkingia* was less prevalent in crops than in midguts. We did find that two single midgut samples lacked *Elizabethkingia* and clustered separately from the other midgut samples (but also separately from crops). This is not unexpected, as single samples will reveal aspects of bacterial communities that are clouded by pooling, especially in the case of overdominant taxa, but it does serve to underscore the value of sequencing non-pooled samples when individual variation is of interest.

Gusmão et al. [20] cultured bacteria from laboratory reared *Ae. aegypti* adult female crops on Brain Heart Infusion agar. They primarily observed yeast species and *Serratia* in *Ae. aegypti* crops. As recently reviewed by Steven et al (2022) [52], *Serratia* is a relevant member of the microbiota of *Ae. aegypti*. It has been experimentally shown to impact larval survival and vector competence. The mechanisms behind the observed phenotypes are complex and not fully understood given that they may be positive, neutral, or negative depending on the life stage of the host and the pathogen involved in the interactions [52]. We did not identify *Serratia* in our crop or midgut communities, though we did observe the presence of other Enterobacterales (*Klebsiella* and *Enterobacter*) sequences. Additionally, ASV_22 from our study was assigned to Enterobacterales but was not identified at the genus level. Thus, we cannot rule out that *Serratia* is present in our community data set as an unidentified ASV. The fact that neither sequencing nor culture methods revealed the presence of this commensal highlights how spatiotemporal dynamics, and/or strain-specific factors may be at play when we describe the membership of host associated bacterial communities. We also note differences between our findings and those of Guégan et al. [30] who reported no significant differences in alpha diversity and bacterial community structures when comparing crop and midgut samples from laboratory reared *Ae. albopictus*. They found that Weeksellaceae was the most abundant taxa in female crops, and Burkholderiaceae was most dominant in midguts. They did not find any Acetobacteraceae present in any tissue. Differences between their findings and ours may be due to several experimental design differences such as the mosquito species, age, and/or diet of the females. Similarly to our study, the two previous studies of *Aedes* crop microbiota [20,30] were performed in the laboratory, and since mosquitoes primarily acquire their microbiota from the environment [76,80], laboratory rearing inherently influences and limits the potential members of the microbiota to those present in the artificial laboratory environment. This may explain the lack of Acetobacteraceae in crops of *Ae. albopictus*. It would be valuable to assay the crop and midgut microbiotas of field-collected mosquitoes to determine if

Acetobacteraceae is common in the crops of wild mosquitoes. Wild mosquitoes will be exposed to a different suite of bacteria in their natural larval development sites and will feed on nectar as adults instead of pure sucrose which is not only nutritionally different but may also contain antimicrobial compounds that can influence the crop bacterial community [81].

One possible driver of the differences we observed in microbial communities between the crop and midgut may be mosquito physiology. Indeed, we note reports of bacterial community structures specific to mosquito organs such as midgut, salivary glands, ovaries, and Malpighian tubules, suggesting these tissues may reflect distinct niches in the mosquito [49,82–84]. One potential physiological difference that may differ between the crop and midgut is pH. The adult midgut has been shown to be actively maintained at slightly acidic conditions in sugar fed females (approximately pH 6.0) while the crop does not regulate the pH of its contents [85]. Therefore, if acid-producing bacteria colonize the crop, it may result in acidification which could influence growth of bacteria that cannot tolerate low pH. Gusmão et al. [20] found a dominant *Serratia* sp. growing in crops with the ability to produce acid when in the presence of glucose. In parallel, they observed acidification of crop contents over the course of 24 hours. In our study, *Tanticharoenia*, which is abundant in the crop, has the capacity to secrete acetic acid during fermentation. Notably, fermentation is an enriched functional process in the crop-associated bacterial community (Fig 8). We did not measure pH of the crop or midgut in our study; nonetheless, it is not unreasonable to hypothesize that AAB could be influencing bacterial diversity in the crop by altering the pH, while in the midgut, active regulation of the pH could limit the impact of acid producing bacteria and result in a different outcome in the community assembly process.

Microbial community structures may also be determined by the mosquitoes' organ-specific immune responses. In the midgut, immune system signaling has been shown to control proliferation of bacteria [86–88]. Homeostasis within this complex system is regulated in part by a mechanism involving mosquito C-type lectins used by commensal bacteria to evade the effects of antimicrobial peptides (AMPs) [89,90]. To the best of our knowledge, the first study characterizing immune gene transcript abundance in the *Ae. aegypti* crop was recently published by Hixson et al. [91]. The authors propose that the predominant phenomenon in the crop would be immune recognition via orthologs of the Toll pathway. Similar to the posterior and hind gut, low abundance of gambicin and lyzozyme transcripts were detected in the crop. Segments with this transcriptional profile are hypothesized by the authors to be areas of microbial tolerance rather than bacterial community modulation as would be predicted in tissues with high AMP transcript abundance such as the midgut.

It is also possible that the differences in bacterial community structures between the crop and midgut were driven by bacteria-bacteria interactions within each niche. Bacteria interact metabolically with neighboring community members within short ranges [92]. The presence of a single nutrient source (e.g. carbohydrates) in a microenvironment can lead to bacterial exclusion and communities that are uneven and have low diversity [93]. This community dynamics model provides a plausible explanation for what we observed in our study. Alternatively, the dominance of *Elizabethkingia* in midguts could derive from antagonistic bacteria-bacteria interactions. It was recently reported that when co-cultured, *Elizabethkingia anophelis* inhibits the growth of a *Pseudomonas* sp. through an antimicrobial-independent mechanism [94]. In addition, the genome of *Elizabethkingia meningoseptica* contains genes predicted to confer resistance to multiple antibiotics [94–96] and other toxic compounds [96], potentially conferring advantage over other commensals when in hostile environments.

Many of the dominant bacteria in the crops (namely the Acetobacteraceae) were also found in the midgut tissues, albeit in smaller proportions and amounts than in the crops. This may indicate that some bacteria are transferred between these tissues. An example of this may

occur if bacteria accompany the sugar that is pumped from the crop into the midgut when the mosquito needs to absorb its sugar reserves. However, it is possible that the proventriculus prevents most bacteria from moving from the foregut into the midgut as is seen with other insects [97]. Lanan et al. [97] demonstrated in the ant *Cephalotes rhoweri* that the proventriculus acts as a bacterial filter and has a similar bacterial structure to the crop. Because our midgut samples included the proventriculus, the small amounts of crop-associated bacteria found in midguts may actually represent bacteria existing in the proventriculus. As bacteria transit from the mosquito crop to the midgut, gambicin and lysozyme produced in the proventriculus may exert selective pressure. In this scenario, the host's immune system would be shaping the seeding community of the posterior gut segments [91].

Spatiotemporal dynamics may also be a critical determinant of community assembly along the digestive tract. As colonizing microbes enter the community either through internal transstadial transmission or ingestion of water and/or nectar [6,8,9,98], the invading order of each species would determine the community structure at each habitat. This phenomenon, known as the multiple stable states [99,100] has been tested in a *Drosophila* gut model [101]. This study revealed how, as each habitat along the gut faces the influx of new species, stochasticity and bottlenecks induce (probabilistic) hysteresis in the colonization process. Though deterministic processes (mediated by tissue attachment) could be at work, both crop and midgut communities would be unstable habitats in the fly model as they are colonized by bacteria in the lumen [101]. Studies under this ecological framework are yet to be performed in mosquito models, but community ecology approaches are highly relevant for understanding mosquito-microbiome interactions [52].

No functional studies have been performed, to the best of our knowledge, characterizing the role bacteria may play in the crop's metabolic pathways. David et al. [65] report a functional analysis of midgut samples from sugar fed females using PICRUST. Though the functional categories are different to those in FAPROTAX, both approaches suggest the potential of the midgut bacterial community to perform carbon, nitrogen, and energy metabolism. Consistently with other models [102,103], we observed that the functional trait abundances of the bacterial communities reflect the overall beta diversity, as samples cluster predominantly in a by-tissue pattern (Fig 8). We detected a significant difference in methanol oxidation potential between the crop and midgut. Methanol catabolizing bacteria are methylotrophs and methanotrophs that establish and thrive when low pH acts as a niche-defining factor [104,105]. These organisms develop in the presence of plant derived sugars, being predominant members of the Earth's phyllosphere [104,105]. The community enriched functions we report could be attributed to crop exclusive methylotrophs like *Methylobacterium* and *Methylovirgula* present in low abundances in the bacterial assembly. While we do not know whether methanol detoxification is a relevant phenomenon for adult mosquitos, it has been reported to be for mosquito larvae [106] and other insects that feed on plants [107].

Given that commensal community structures are influenced by deterministic and neutral processes when colonizing a niche along an insect's gut, spatiotemporal scales and hysteresis in the community assembly process have to be considered [101]. As such, we acknowledge that the community structures we report for each organ are community snapshots given particular physiological host states, contextualized by the sex, age and nutritional states of the mosquitos we used. Nevertheless, our study demonstrates that different sections of the mosquito digestive tract can contain dramatically different communities of bacteria. Most notably, our study showed that the mosquito midgut is often overwhelmingly inhabited by *Elizabethkingia*, while the crop is generally more diverse and has a higher proportion of *Asaia* and *Tanticharoenia* bacteria. These distinct bacterial profiles may confer different biological functions, such as altering pH or metabolizing particular compounds. It is likely that multiple factors

influence the development of these communities, such as host intrinsic parameters, bacteria-bacteria interactions, exchange of bacteria between tissues, and stochastic effects. Understanding how these communities assemble may facilitate better methods to engineer a mosquito microbiome that is refractory to pathogen transmission with the ultimate goal of reducing the disease burden that mosquitoes, especially *Ae. aegypti*, pose to people around the world [49,52, 108]. In addition, future work investigating organ-specific microbial communities has the potential to elucidate how production and exchange of secondary metabolites can influence within-host bacterial community dynamics and host health as seen in other models [109].

## Supporting information

**S1 Fig. Histogram showing read counts for all samples.**
(TIF)

**S2 Fig. PCoA for all samples including buffer blanks.** PCoA using Bray Curtis dissimilarity values shows that buffer blanks (shown in green) group together and separately from experimental samples.
(TIF)

**S3 Fig. Rarefaction curves for all sequencing samples.**
(TIF)

**S1 File. Raw data file.** Each tab in the spreadsheet corresponds to a different figure or subfigure.
(XLSX)

**S2 File. R code used for sequence processing, downstream bioinformatics, and statistical analyses for high-throughput 16S amplicon sequence data.**
(R)

## Acknowledgments

We wish to acknowledge the insectary staff at the Johns Hopkins Bloomberg School of Public Health for assistance with mosquito rearing.

## Author Contributions

**Conceptualization:** Sarah M. Short.

**Data curation:** Luis E. Martinez Villegas, James Radl, Sarah M. Short.

**Formal analysis:** Luis E. Martinez Villegas, James Radl, Sarah M. Short.

**Funding acquisition:** George Dimopoulos, Sarah M. Short.

**Investigation:** Luis E. Martinez Villegas, James Radl, Sarah M. Short.

**Methodology:** Sarah M. Short.

**Supervision:** George Dimopoulos, Sarah M. Short.

**Visualization:** Luis E. Martinez Villegas, James Radl, Sarah M. Short.

**Writing – original draft:** Luis E. Martinez Villegas, James Radl, Sarah M. Short.

**Writing – review & editing:** Luis E. Martinez Villegas, James Radl, George Dimopoulos, Sarah M. Short.

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
