## [Decision Letter · Decision Letter 0]

14 Nov 2022

Dear Dr Sarah M. Short,

Thank you very much for submitting your manuscript "Bacterial communities of Aedes aegypti mosquitoes differ between crop and midgut tissues" for consideration at PLOS Neglected Tropical Diseases. As with all papers reviewed by the journal, your manuscript was reviewed by members of the editorial board and by several independent reviewers. The reviewers appreciated the attention to an important topic. Based on the reviews, we are likely to accept this manuscript for publication, providing that you modify the manuscript according to the review recommendations. 

The reviewers and I suggest a minor revision of the manuscript. Based on the two reviews, this manuscript for publication should be accepted for publication, providing that authors modify the manuscript according to the review recommendations. Please note. comments from reviewer 2 are provided un attached file.

Sincerely,

Katja Fischer

Academic Editor

Alvaro Acosta-Serrano

Section Editor

The reviewers and the Academic Editor suggest a minor revision of the manuscript. Based on the two reviews, this manuscript for publication should be accepted for publication, providing that authors modify the manuscript according to the review recommendations. Please note. comments from reviewer 2 are provided as attached file.

Reviewer's Responses to Questions

**Key Review Criteria Required for Acceptance?**

**Methods**

-Are the objectives of the study clearly articulated with a clear testable hypothesis stated?

-Is the study design appropriate to address the stated objectives?

-Is the population clearly described and appropriate for the hypothesis being tested?

-Is the sample size sufficient to ensure adequate power to address the hypothesis being tested?

-Were correct statistical analysis used to support conclusions?

-Are there concerns about ethical or regulatory requirements being met?

Reviewer #1: When studying potential impact of microbiota in mosquitoes’ biological function, it may be more relavent to investigate the microbiota composition of individual mosquitoes. Please provide rationale for using pooled vs individual samples.

Provide rationale for the use of TSA and M9 media. Is there anticipation to see any differences in the types of colonies grown between these two media. If so, please include this in the discussion.

Reviewer #2: (No Response)

**Results**

-Does the analysis presented match the analysis plan?

-Are the results clearly and completely presented?

-Are the figures (Tables, Images) of sufficient quality for clarity?

Reviewer #1: Was there any sequences of animal parasites or symbionts detected in the 16S sequencing? Metabolic profiles 10-20% heat map in Fig 8 seems to indicate, this may be the case in some samples, in both crops and midguts.

Reviewer #2: (No Response)

**Conclusions**

-Are the conclusions supported by the data presented?

-Are the limitations of analysis clearly described?

-Do the authors discuss how these data can be helpful to advance our understanding of the topic under study?

-Is public health relevance addressed?

Reviewer #1: Line 480-484- It is stated that presence or absence of Elizabethkingia may influence the size of microbiota in the midguts (individual samples). Please clarify in the result sections that samples 4 and 7 (Fig 4) represent the two bottom outliners in (Fig 2).

Please discuss potential role of Serratia in Ae. aegypti.

Discuss the differences in the findings between the pooled vs individual samples.

Reviewer #2: (No Response)

**Editorial and Data Presentation Modifications?**

Reviewer #1: No concerns.

Reviewer #2: (No Response)

**Summary and General Comments**

Reviewer #1: This study investigates the microbial composition of crops vs midguts in Aedes aegypti mosquitoes, which are important vector for various viral diseases. Authors utlised both culture-dependent and culture independent (16S rDNA) approach to determine the community compositions in individual and pooled samples of Ae. aegypti. Tanticharoenia and Asaia dominated crops while Eliazbethkingia dominated mid gut. These findings differ from another similar study, however may be due to different experimental conditions.

In the introduction: 

It will be beneficial to include why the study focused only on the female Ae aegypti rather than both males and females and a figure/drawing of mosquito’s gut anatomy. For clarity, please state whether the mosquitoes are lab reared or field samples when referring to other studies.

Reviewer #2: (No Response)

PLOS authors have the option to publish the peer review history of their article (what does this mean?). If published, this will include your full peer review and any attached files.

Reviewer #1: No

Reviewer #2: No

Figure Files:

Data Requirements:

Reproducibility:

References

---

## [Decision Letter · Decision Letter 1]

6 Mar 2023

Dear Authors,

We are pleased to inform you that your manuscript 'Bacterial communities of Aedes aegypti mosquitoes differ between crop and midgut tissues' has been provisionally accepted for publication in PLOS Neglected Tropical Diseases.

Best regards,

Katja Fischer

Academic Editor

Alvaro Acosta-Serrano

Section Editor

Reviewer's Responses to Questions

**Key Review Criteria Required for Acceptance?**

**Methods**

-Are the objectives of the study clearly articulated with a clear testable hypothesis stated?

-Is the study design appropriate to address the stated objectives?

-Is the population clearly described and appropriate for the hypothesis being tested?

-Is the sample size sufficient to ensure adequate power to address the hypothesis being tested?

-Were correct statistical analysis used to support conclusions?

-Are there concerns about ethical or regulatory requirements being met?

Reviewer #1: (No Response)

Reviewer #2: (No Response)

**Results**

-Does the analysis presented match the analysis plan?

-Are the results clearly and completely presented?

-Are the figures (Tables, Images) of sufficient quality for clarity?

Reviewer #1: (No Response)

Reviewer #2: (No Response)

**Conclusions**

-Are the conclusions supported by the data presented?

-Are the limitations of analysis clearly described?

-Do the authors discuss how these data can be helpful to advance our understanding of the topic under study?

-Is public health relevance addressed?

Reviewer #1: (No Response)

Reviewer #2: (No Response)

**Editorial and Data Presentation Modifications?**

Reviewer #1: (No Response)

Reviewer #2: (No Response)

**Summary and General Comments**

Reviewer #1: Authors have addressed previous review comments. Recommend acceptance.

Reviewer #2: My suggested revisions were addressed, I have no additional comments.

PLOS authors have the option to publish the peer review history of their article (what does this mean?). If published, this will include your full peer review and any attached files.

Reviewer #1: No

Reviewer #2: No

---

## [Editor Report · Acceptance letter]

24 Mar 2023

Dear Dr Short,

We are delighted to inform you that your manuscript, "Bacterial communities of *Aedes aegypti* mosquitoes differ between crop and midgut tissues," has been formally accepted for publication in PLOS Neglected Tropical Diseases.

Best regards,

Shaden Kamhawi

co-Editor-in-Chief

Paul Brindley

co-Editor-in-Chief
